# A Novel Near-Infrared Ytterbium Complex [Yb(DPPDA)_2_](DIPEA) with *Φ* = 0.46% and *τ_obs_* = 105 μs

**DOI:** 10.3390/molecules28041632

**Published:** 2023-02-08

**Authors:** Guozhu Ren, Danyang Zhang, Hao Wang, Xiaofang Li, Ruiping Deng, Shihong Zhou, Long Tian, Liang Zhou

**Affiliations:** 1State Key Laboratory of Rare Earth Resource Utilization, Changchun Institute of Applied Chemistry, Chinese Academy of Sciences, Changchun 130022, China; 2School of Applied Chemistry and Engineering, University of Science and Technology of China, Hefei 230027, China; 3School of Materials Science and Engineering, Jilin Jianzhu University, Changchun 130118, China

**Keywords:** ytterbium complex, near-infrared luminescence, absolute quantum yield, luminescent lifetime, intrinsic quantum yield, internal redox mechanism

## Abstract

The luminescent performances of near-infrared (NIR) lanthanide (Ln) complexes were restricted greatly by vibration quenching of X-H (X = C, N, O) oscillators, which are usually contained in ligands and solvents. Encapsulating Ln^3+^ into a cavity of coordination atoms is a feasible method of alleviating this quenching effect. In this work, a novel ytterbium complex [Yb(DPPDA)_2_](DIPEA) coordinated with 4,7-diphenyl-1,10-phenanthroline-2,9-dicarboxylic acid (DPPDA) was synthesized and characterized by FT-IR, ESI-MS and elemental analysis. Under the excitation of 335 nm light, [Yb(DPPDA)_2_](DIPEA) showed two emission peaks at 975 and 1011 nm, respectively, which were assigned to the characteristic ^2^F_5/2_ → ^2^F_7/2_ transition of Yb^3+^. Meanwhile, this ytterbium complex exhibited a plausible absolute quantum yield of 0.46% and a luminescent lifetime of 105 μs in CD_3_OD solution. In particular, its intrinsic quantum yield was calculated to be 12.5%, and this considerably high value was attributed to the near-zero solvent molecules bound to Yb^3+^ and the absence of X-H oscillators in the first coordination sphere. Based on experimental results, we further proposed that the sensitized luminescence of [Yb(DPPDA)_2_](DIPEA) occurred via an internal redox mechanism instead of an energy transfer process.

## 1. Introduction

Lanthanide (Ln) ions have been extensively studied in recent years for their characteristic features such as sharp emission spectrum, long luminescent lifetime and wide spectroscopic range [1]. However, lanthanide ions possess low molar absorption coefficients due to their parity-forbidden *4f-4f* transitions [2]. In 1941, Weissman reported that the fluorescence of Eu^3+^ could be sensitized by a chromophore acting as an antenna for light absorption [3,4]. Since then, a number of complexes containing various ligands with large extinction coefficients and appropriate excited state energies have been synthesized and applied to improve the luminescent performances of lanthanide ions [5,6].

In particular, near-infrared (NIR) luminescent complexes centered at Nd^3+^, Er^3+^ and Yb^3+^ have attracted increasing attention because of their tremendous potential for application in telecommunication and bio-imaging fields [7,8,9]. Nevertheless, NIR lanthanide complexes suffer from low intrinsic quantum yields and short luminescent lifetimes because radiative transitions of Nd^3+^, Er^3+^ and Yb^3+^ could be easily quenched by high energy vibration of X-H (X = C, H, O) oscillators, which are usually contained in organic ligands, solvent molecules and moisture [10]. Therefore, some methods were adopted to diminish this impact on NIR luminescent properties such as replacing C-H bonds with lower vibrational energy C-F and C-D oscillators or by protecting the lanthanide ions from deleterious solvent/moisture coordination through structural adjustment [11,12,13,14].

With regard to the difficulty and high cost of the perdeuteration and perhalogenation process, NIR lanthanide complexes with non-deuterated and non-halogenated ligands, which are relatively convenient to synthesize and exhibit considerable luminescent efficiency, have been reported frequently in recent years [15,16,17,18,19,20]. In these works, plenty of efforts were devoted to making Ln^3+^ surrounded by coordination atoms in the state of a spherical shell. As a result, the solvent molecules could be blocked out of the first coordination sphere, which would then alleviate the quenching effect. Consequently, the quantum yields (QYs) and luminescent lifetimes (τ) of the complexes could be elevated. However, as far as we know, only a handful of ytterbium complexes could reach 3% QY and 72 μs τ in deuterated solvents until now [15,19].

Herein, we prepared a novel ytterbium complex with non-deuterated and non-halogenated ligand 4,7-diphenyl-1,10-phenanthroline-2,9-dicarboxylic acid (DPPDA). Moreover, a nonnucleophilic base N-ethyl-N,N-diisopropylamine (DIPEA) was adopted to facilitate the coordination reaction and balance the charge of target complexes. The synthesized [Yb(DPPDA)_2_](DIPEA) showed a plausible absolute luminescent QY of 0.46% and long τ of 105 μs in CD_3_OD solution at room temperature. Surprisingly, the intrinsic QY of the complex was 12.5% and was rather high for such species of ytterbium complexes. In the end, we proposed that the sensitized luminescence of [Yb(DPPDA)_2_](DIPEA) occurred via an internal redox mechanism rather than an energy transfer process.

## 2. Results and Discussion

The synthetic routes of the ligand DPPDA and the Ln (Ln = Yb^3+^, Gd^3+^) complexes [Ln(DPPDA)_2_](DIPEA) are depicted in Figure 1. The 2,9-Dimethyl-4,7-diphenylphenanthroline was oxidized to gain 2,9-Dicarbaldehyde-4,7-diphenylphenanthroline in the presence of SeO_2_ and then further oxidized to obtain DPPDA with 48% HNO_3_. Afterwards, the DPPDA was coordinated to Ln^3+^ catalyzed by DIPEA at room temperature. ^1^H-NMR, ^13^C-NMR, FT-IR, ESI-MS and elemental analysis (Figure 1 and Appendix A (Appendix A)) were carried out to determine if the syntheses of the ligand and complexes were successful. ^1^H-NMR and ^13^C-NMR spectra of Ln complexes were not obtained due to the paramagnetism of Yb^3+^ and Gd^3+^.

### 2.1. FT-IR Analysis

The FT-IR spectra of DPPDA and [Yb(DPPDA)_2_](DIPEA) in the 1800~1350 cm^−1^ range are shown in Figure 1 and the full spectra in the 4000~400 cm^−1^ range are presented in Appendix A. The broad absorption bands between 3700 and 2200 cm^−1^ are ascribed to O-H stretching of DPPDA and N-H stretching of (DIPEA·H)^+^ in [Yb(DPPDA)_2_](DIPEA), respectively. The peak of the ligand at 1740 cm^−1^, which is assigned to C=O stretching of carboxyl in DPPDA, disappears while in the complex because of the formation of carboxylate. Moreover, there are two vibration coupling peaks (1648 and 1408 cm^−1^) of the carboxylate appearing in the spectrum of the complex. In addition, the infrared absorption peaks in the range of 1620~1575, 1505~1490 and 1455~1445 cm^−1^ are attributed to C=C stretching of the aromatic ring and peaks in the range of 1560~1545 cm^−1^ are assigned to C=N stretching of DPPDA. Moreover, a general shift of ring vibration in the range of 1620~1440 cm^−1^ to high frequencies for [Yb(DPPDA)_2_](DIPEA) compared with ligand confirmed that DPPDA had been successfully coordinated to Yb^3+^ [21].

### 2.2. UV-Vis Analysis

The UV-Vis absorption spectra of DPPDA and [Ln(DPPDA)_2_](DIPEA) are depicted in Figure 2. There are three absorption peaks for each of them, which can be attributed to the spin-allowed π → π* and spin-forbidden n → π* electronic transitions of DPPDA. Furthermore, the absorption peaks of [Ln(DPPDA)_2_](DIPEA) (240 nm, 296 nm, 327 nm) are red-shifted to different extents compared with those of DPPDA (236 nm, 285 nm, 324 nm). This result demonstrated that the conjugated degree of ligands had been enlarged in complexes and Ln(III) ions were successfully coordinated with DPPDA.

### 2.3. Thermal Stability Characterization

The TG and DTG curves of [Yb(DPPDA)_2_](DIPEA) are provided in Figure 3. The weight loss below 150 °C is due to the decomposition of crystalline water molecules in the complex. More importantly, [Yb(DPPDA)_2_](DIPEA) is thermostable below 300 °C, which is quite excellent for lanthanide complexes. When the temperature increased persistently, 15.6% and 11.8% weight loss steps appeared, which are consistent with the CO_2_ loss caused by decarboxylation (15.4%) and the loss arising from deprivation of DIPEA (11.3%), respectively. The superior thermal stability of this complex indicates that it might be suitable for telecommunication and display applications.

### 2.4. Luminescent Properties

The excitation and emission spectra of the complex [Yb(DPPDA)_2_](DIPEA) are rendered in Figure 4a. The excitation band ranges from 242 to 378 nm and peaks at 335 nm when monitored at 1011 nm light. Under the excitation of a 335 nm light, [Yb(DPPDA)_2_](DIPEA) shows two narrow NIR emission bands at 975 and 1011 nm, respectively, which are assigned to the characteristic ^2^F_5/2_ → ^2^F_7/2_ transition of Yb^3+^. According to Appendix A, we found that the excitation spectrum of [Yb(DPPDA)_2_](DIPEA) significantly overlapped with the UV-Vis absorption spectrum of the complex. This reveals that the energy absorbed by DPPDA could be utilized to stimulate the NIR luminescence of Yb^3+^. In addition, the emission spectra of [Yb(DPPDA)_2_](DIPEA) and [Gd(DPPDA)_2_](DIPEA) within the UV-Vis zone, which arose from the radiative transition of DPPDA, were obtained under the same testing conditions (excitation light wavelength, light source slit, detector slit and concentration of solutions), and are shown in Figure 4b. Since the excited state energies of Gd^3+^ (higher than 31,000 cm^−1^) are quite high compared to Yb^3+^ (^2^F_5/2_, 10,235 cm^−1^), it is difficult for Gd^3+^ to accept the energy transferred from DPPDA. Consequently, the ligand-centered emission of [Gd(DPPDA)_2_](DIPEA) is considerably stronger compared with the visible emission of [Yb(DPPDA)_2_](DIPEA). In addition, the significantly diminished visible luminescent intensity of the ytterbium complex indicates that the energy absorbed by DPPDA has been plentifully transferred to ytterbium ions. However, it is worth noting that [Yb(DPPDA)_2_](DIPEA) shows residual UV-Vis fluorescent emission, which is attributed to singlet radiative transition of the ligand, having roughly 21% relative intensity compared with that of [Gd(DPPDA)_2_](DIPEA). Consequently, we can speculate that the energy transfer in [Yb(DPPDA)_2_](DIPEA) is incomplete and there is still room for further improvement in energy transfer efficiency. In summary, we conclude that the energy absorbed by DPPDA has been transferred to Yb^3+^ [15,22], which confirms the potential of DPPDA to sensitize the luminescence of Yb^3+^.

The luminescence decay curves of [Yb(DPPDA)_2_](DIPEA) in different solutions and solid state are provided in Appendix A. All the decay profiles are single exponential functions, which point to the presence of only one emissive ytterbium center [23]. The luminescent lifetimes (*τ_obs_*) of [Yb(DPPDA)_2_](DIPEA) are calculated and summarized in Table 1. Surprisingly, the *τ_obs_* of complex in CD_3_OD (105 μs) has until now, as far as we know, the longest luminescent lifetime of non-deuterated and non-halogenated ytterbium complexes [13,14,16,20]. Moreover, according to the literature [16], the luminescent lifetimes of deuterated and non-deuterated solvents can be used to quantify the number of solvent molecules coordinated with the lanthanide cations. The calculation formula for the solvation number is shown in the following equation:(1)$$q=A(k_{H}-k_{D})-B$$

Here, *q* is the number of solvent molecules bound to lanthanide ions in the first sphere of coordination, *A* is a constant related to the sensitivity of Ln^3+^ to high vibrational energy quenching (*A* = 2 μs for Yb^3+^ in CH_3_OH), *k_H_* = 1/τ_H_ (τ_H_ is the *τ_obs_* in CH_3_OH), *k_D_* = 1/τ_D_ (τ_D_ is the *τ_obs_* in CD_3_OD) and *B* is a correction factor (*B* = 0.1 for Yb^3+^ in CH_3_OH) [24,25]. The value of q is calculated to be 0.08 and this near-zero value reveals that there are no solvent molecules bound to Yb^3+^ in its first coordination sphere. Moreover, this means that ytterbium ions have been effectively protected from the deleterious solvent molecules, which made it possible to promote the quantum yield and luminescent lifetime. In addition, there are no C-H oscillators in the first coordination sphere of Yb^3+^ and it is absolutely beneficial for the improvement of luminescent performances.

The absolute quantum yields of [Yb(DPPDA)_2_](DIPEA) in different solvents are listed in Table 1. Detailed measurement data are presented in Appendix A. The absolute quantum yields are quite low, and the luminescent lifetimes are slightly long for ytterbium complexes with non-deuterated and non-halogenated ligands. Interestingly, the significant deterioration in the performance of [Yb(DPPDA)_2_](DIPEA) in CH_3_OH compared with CHCl_3_ is probably due to the smaller size and closer distance to the Yb^3+^ of CH_3_OH than CHCl_3_, which results in a faster quenching rate of [Yb(DPPDA)_2_](DIPEA) in CH_3_OH. In addition, among these results [Yb(DPPDA)_2_](DIPEA) showed 0.46% quantum yield and 105 μs lifetime in X-H-free CD_3_OD solution. The comparatively low QYs were possibly related to the impurity of [Yb(DPPDA)_2_](DIPEA) because the crude products were purified by rough recrystallization rather than a precise sublimation process. Based on the literature [26], to obtain more insight into the sensitized luminescent process, the refractive index (n = 1.3271) and NIR absorption spectrum (shown in Appendix A) of [Yb(DPPDA)_2_](DIPEA) in 4 × 10^−4^ mol/L CD_3_OD solution were measured to determine the *τ_rad_* of the complex. Furthermore, a series of luminescent properties such as intrinsic quantum yield ($\Phi_{Ln}^{Ln}$) and sensitization efficiency (*η_sens_*) were calculated using the following equation:(2)$$\phi_{Ln}^{L}=\eta_{sens}\phi_{Ln}^{Ln}=\eta_{sens}\frac{\tau_{obs}}{\tau_{rad}}$$

Then, the full evaluation of [Yb(DPPDA)_2_](DIPEA) could be obtained and is presented in Table 2. The value of $\Phi_{Ln}^{Ln}$ is considerably high for ytterbium complexes, which means that X-H oscillators contained in solvent molecules and ligands have been blocked out of the first coordination sphere of Yb^3+^, so the vibration quenching effect could be alleviated. However, the value of *η_sens_* is rather low, which is related to the inefficiency of Yb^3+^ in utilizing the energy transferred from DPPDA and is caused by some non-radiative competitive processes. Additionally, it is essential to further optimize the structure of ligand to enhance the sensitization efficiency in follow-up works.

### 2.5. Luminescent Mechanisms

The conventional energy transfer mechanism of sensitized luminescence of Yb (III) is depicted in Figure 5. The main energy migration path includes spin-allowed ligand-centered absorption followed by intersystem crossing (^1^S_1_* → ^3^T*), energy transfer process (^3^T* → Yb^3+^*) and Yb^3+^ emission. It is also necessary that the triplet state of ligand significantly overlaps with the absorption band of Yb^3+^. To precisely determine the degree of energy matching between ^3^T* and Yb^3+^* in [Yb(DPPDA)_2_](DIPEA), the low-temperature fluorescence and phosphorescence spectra of [Gd(DPPDA)_2_](DIPEA) were measured (as shown in Appendix A). The positions of ligand-based singlet and triplet states are 19,841 and 19,646 cm^−1^, respectively, and dramatically mismatch with the energy of ^2^F_5/2_ (10,235 cm^−1^). With regard to the requirement of a 2500 to 3500 cm^−1^ energy difference between ^3^T* and Yb^3+^* for efficient sensitization [2], the energy transfer mechanism is not suitable for interpreting the sensitized luminescence of [Yb(DPPDA)_2_](DIPEA).

Considering how the electron configuration of Yb^3+^ (4f^13^) tends to accept one electron and become more stable (4f^14^), Yb^3+^ is slightly easier to reduce to Yb^2+^. At the same time, ligand DPPDA^2−^ is quite electron-rich, so it is possible for DPPDA^2−^ to lose one electron and become DPPDA^−.^. Therefore, we hypothesize that the sensitized luminescence of Yb^3+^ occurs via an internal redox process [15,27,28], as shown in Figure 6. Yb^3+^ will be stimulated to an excited state indirectly via ligand-centered absorption (DPPDA^2−^ → DPPDA^2−*^) as well as forward electron transfer (DPPDA^2−*^, Yb^3+^ → DPPDA**^−.^**, Yb^2+^) and back electron transfer (DPPDA**^−.^**, Yb^2+^ → DPPDA^2−^, Yb^3+*^) processes. Meaningfully, it is necessary to contrast the energy of all excited states to determine whether or not there is a thermodynamic driving force among these processes. According to the literature [28], the driving force ($-{\Delta G}_{\mathrm{Yb}}$) in the forward electron transfer process can be estimated using the following equation:(3)$${\Delta G}_{\mathrm{Yb}}=E\left({\mathrm{DPPDA}}^{-.}/{\mathrm{DPPDA}}^{2-}\right)-E\left({\mathrm{DPPDA}}^{2-*}\right)-E\left({\mathrm{Yb}}^{3+}/{\mathrm{Yb}}^{2+}\right)$$

Here, the lowest excited energy state of DPPDA^2−^ E(DPPDA^2−*^) is calculated to be 3.40 eV (shown in Figure 4a) and the electrode potential of Yb^3+^/Yb^2+^ E(Yb^3+^/Yb^2+^) is known to be *−*1.05 eV [28]. At the same time, the electrode potential of DPPDA^−.^/DPPDA^2−^ E(DPPDA^−.^/DPPDA^2−^) is demonstrated to be 1.25 eV by cyclic voltammogram, which is shown in Appendix A (1.03 eV plus Ag/AgCl reference electrode potential 0.22 eV). Subsequently, the driving force in the forward electron transfer process is estimated to be 1.10 eV and the energy of ‘DPPDA**^−.^**, Yb^2+^’ is calculated to be 2.30 eV. Consequently, it is theoretically feasible for DPPDA^2−^ to sensitize the luminescence of Yb^3+^ through the internal redox process and the comparatively low sensitization efficiency is probably due to the competition between two back electron-transfer processes including ‘DPPDA**^−.^**, Yb^2+^ → DPPDA^2−^, Yb^3+*^’ and ‘DPPDA**^−.^**, Yb^2+^ → DPPDA^2−^, Yb^3+^’. This issue is interesting and important for the design and optimization of rare earth complexes; as such, we will carry out further experimental investigation to verify the internal redox mechanism in our next work.

## 3. Materials and Methods

### 3.1. Drugs and Reagents

All reagents were used as received unless otherwise stated: 2,9-Dimethyl-4,7-diphenylphenanthroline (Aladdin Co., Shanghai, China); selenium(IV) oxide (Aladdin Co., China); 1,4-dioxane; concentrated nitric acid; acetonitrile; anhydrous ethanol; ytterbium(III) chloride (Aladdin Co., China); gadolinium(III) chloride (Macklin Co., China); *N*-ethyl-*N*,*N*-diisopropylamine (Alfa Aesar Co., China); and Tetrabutylammonium perchlorate (Energy Co., Shanghai, China). All chemicals were analytical-grade reagents.

### 3.2. Instruments

^1^H-NMR and ^13^C-NMR spectra were obtained from the AVANCE III HD 500 liquid high-resolution superconducting NMR spectrometer. FT-IR spectra were measured within a 4000*–*400 cm^−1^ region on a VERTEX70 Fourier transform infrared spectrometer (BRUKER, Bremen, Germany) using the KBr pellet technique. ESI-MS spectra were recorded on the Quattro Premier XE mass spectrometer (Waters, Milford, MA, USA). Element analysis was carried out on a Vario EL cube elemental analyzer. UV-Vis absorption spectra were obtained from a TU-1901 UV-Vis spectrometer (PERSEE, Beijing, China). The TG curve was obtained from a TGA/DSC 1/1100 thermogravimetric analyzer (METTLER, Greifensee, Switzerland). The photoluminescent excitation and emission spectra were measured on a FLSP-920 steady-state and time-resolved fluorescence spectrometer (Edinburgh, UK). Luminescence decay curves were collected using an FLSP-920 steady-state and time-resolved fluorescence spectrometer with a μF 920H lamp as the excitation source. UV-Vis emission spectra were carried out on a modular spectrofluorometer equipped with a 450 W xenon lamp as the excitation source (Fluorolog 3; Horiba Jobin Yvon, Longjumeau, France). The NIR absolute quantum yields were measured on an FLS1000 photoluminescence spectrometer equipped with an integrating sphere (Edinburgh, UK). The NIR absorption spectrum was measured on a Lambda 950 UV-Vis-NIR spectrometer (PerkinElmer, Waltham, MA, USA). The refractive index was obtained from an abbe refractometer (LICHEN, Shanghai, China). The low-temperature fluorescence and phosphorescence spectra were obtained from an F-4500 fluorescence spectrophotometer (Hitachi, Tokyo, Japan). The Cyclic voltammogram was measured on a CHI600E electrochemical workstation (Chinstr, Shanghai, China).

### 3.3. Synthesis of 4,7-Diphenyl-1,10-Phenanthroline-2,9-Dicarboxylic Acid (DPPDA)

The synthesis was carried out according to the modified method presented in the literature [29]. The 2,9-Dimethyl-4,7-diphenylphenanthroline (0.465 g, 1.29 mmol) and SeO_2_ (0.674 g, 6.075 mmol) were refluxed in 1,4-dioxane (23 mL)-H_2_O (1 mL) mixed solvent for 3 h under N_2_ atmosphere. After cooling to room temperature, the solvent was removed under reduced pressure and 48% nitric acid (20 mL) was added to the dark brown residue and then refluxed for 3 h. The reaction mixture was cooled to room temperature and poured onto crushed ice. The precipitated light-yellow solid was collected by filtration, washed with water until neutral and finally washed with ice-cold acetonitrile. The obtained products were dried in a vacuum. Yield: 0.46 g (85%). ^1^H-NMR (500 MHz, DMSO) δ 8.23 (s, 2H), 7.96 (s, 2H), 7.53*–*7.70 (m, 10H), 3.31*–*3.97 (br, 2H). ^13^C-NMR (126 MHz, DMSO) δ 166.73, 149.79, 148.52, 146.24, 137.16, 130.19, 129.56, 129.44, 128.12, 126.19, 124.09. IR (KBr disk): 3700*–*2200, 1740, 1615, 1590, 1576, 1546, 1502, 1490, 1446, 1394, 770 and 700 cm^−1^. ESI-MS (CH_3_CH_2_OH negative mode): *m*/*z*: calculated for C_26_H_18_N_2_O_5_·H_2_O, 438.4; found, 438.5. Element analysis: calculated for DPPDA·2H_2_O: C, 68.42%; H, 4.42%; N, 6.14%; O, 21.03%. Found: C, 67.84%; H, 4.30%; N, 6.48%; O, 21.38%.

### 3.4. Synthesis of [Yb(DPPDA)_2_](DIPEA)

DPPDA (1 g, 2.38 mmol) was dissolved in anhydrous ethanol (10 mL) and N-ethyl-N,N-diisopropylamine (DIPEA, 1 mL) was added drop by drop until the light-yellow solid was totally dissolved. Then, ytterbium(III) chloride (0.314 g, 1.13 mmol) dissolved in anhydrous ethanol (5 mL) was mixed with the above solution and the pale-yellow solid was precipitated after stirring at room temperature for 10 h. The resulting mixture was washed with anhydrous ethanol, recrystallized with ethanol, and further dried in a vacuum. Yield: 0.90 g (70%). IR (KBr disk): 3700*–*2200, 1648, 1618, 1598, 1576, 1556, 1504, 1492, 1453, 1408 and 1378 cm^−1^. ESI-MS (CHCl_3_ negative mode): *m*/*z*: calculated for [Yb(DPPDA)_2_]*^−^*, 1010.1; found, 1010.4. ESI-MS (CHCl_3_ positive mode): *m*/*z*: calculated for [[Yb(DPPDA)_2_](DIPEA)+H]^+^, 1141.3; found, 1141.6. Element analysis: calculated for [Yb(DPPDA)_2_](DIPEA)·4H_2_O: C, 59.40%; H, 4.62%; N, 5.77%. Found: C, 59.61%; H, 4.94%; N, 5.52%.

### 3.5. Synthesis of [Gd(DPPDA)_2_](DIPEA)

The synthetic procedure was similar to the method used for [Yb(DPPDA)_2_](DIPEA). DPPDA (0.1 g, 0.238 mmol) was dissolved in anhydrous ethanol (2 mL) and N-ethyl-N,N-diisopropylamine (DIPEA) was added drop by drop until the light-yellow solid was totally dissolved. Then, gadolinium(III) chloride (0.032 g, 0.121 mmol) dissolved in anhydrous ethanol (1.5 mL) was mixed with the above solution and the pale-yellow solid was precipitated. The resulting mixture was washed with anhydrous ethanol and further dried in a vacuum. IR (KBr disk, Appendix A): 3700*–*2200, 1641, 1616, 1595, 1576, 1552, 1504, 1491, 1453 and 1409 cm^−1^. ESI-MS (CHCl_3_ negative mode): *m*/*z*: calculated for [Gd(DPPDA)_2_]*^−^*, 994.1; found, 994.7. ESI-MS (CHCl_3_ positive mode): *m*/*z*: calculated for (DIPEA+H^+^), 130.2; found, 130.0.

### 3.6. Method of Cyclic Voltammetry Testing

DPPDA (8.4 mg, 0.007 mmol), DIPEA (7 μL) and tetrabutylammonium perchlorate (1.36 g, 3.99 mmol, electrolyte) were dissolved in CHCl_3_ (20 mL) to prepare the test solution. The electrochemical properties were investigated using a glass-carbon working electrode, a platinum counter electrode and an Ag/AgCl reference electrode. The test solution was degassed with argon for ten minutes and the cyclic voltammetry scanning speed was set to 50 mV/s.

## 4. Conclusions

In conclusion, we have successfully synthesized a novel near-infrared ytterbium complex [Yb(DPPDA)_2_](DIPEA) with an absolute quantum yield of 0.46% and a luminescent lifetime of 105 μs in CD_3_OD solution. The intrinsic quantum yield of this complex is as high as 12.5%, which is significantly higher than the measured value in CD_3_OD solution because of near-zero solvent molecules bound to Yb^3+^ and the absence of C-H oscillators in the first coordination sphere of Yb^3+^. In addition, the sensitized luminescent mechanism of Yb^3+^ was investigated and, based on our experimental results, the internal redox process was suggested to be the dominant mechanism for the sensitized luminescence of Yb^3+^.

## Data Availability

The data presented in this study are available in the article and the Appendix A.

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
