# Peer review of "A Novel Near-Infrared Ytterbium Complex [Yb(DPPDA)2](DIPEA) with Φ = 0.46% and τobs = 105 μs"

_molecules, 2023, doi:10.3390/molecules28041632_

Round 1

Reviewer 1 Report

In this manuscript, Zhou et al. reported a near-infrared (NIR) lanthanide (Ln) complex, which exhibits absolute quantum yield of 0.46% and luminescent lifetime of 105 μ s in CD3OD solution. The photochemical/physical properties and luminescence mechanism of the complex were investigated in detail. The manuscript is well organized and holds solid evidence in terms of structural characterization. The conclusion is drawn reasonably. Therefore, this work can be accepted and published in Molecules after minor revision. There are some suggestions as followed.
(1) Why does the C=O stretching vibration peak of the carbonyl group disappear after the formation of the complex in FT-IR spectra?

(2) In Figure 2, the author attributes the three absorption peaks of the three compounds to the π-π* electron transition of DPPDA ligand. Are there CT peak or σ-σ* transition?

(3) Why does the complex have the longest luminescent lifetime in deuterated methanol?

(4) Due to the defects of F-4500 fluorescence spectrometer in phosphorescent spectrum testing, the reviewer suggested that the author to use FLS1000 photoluminescence spectrometer for phosphorescent spectrum measurement.

(5) The authors may consider citing other studies on the function of C-F (Adv. Mater., 2022, DOI:10.1002/adma.202208229) and NIR dye (Aggregate, 2021, 2, e59; Angew. Chem. Int. Ed., 2022, 61, e202212673).

(6) The format of references should be carefully checked.

Author Response

Dear reviewer,

        Thank you very much for your meaningful and constructive suggestions. We have made some corrections to our manuscript under consideration of your thoughtful advices. And the responses to your comments have been uploaded in the word file named 'Response to reviewer 1-molecules-2089122' and please see the attachment.

Best wishes

Reviewer 2 Report

The authors presented the synthesis of a novel near-infrared ytterbium complex 3 [Yb(DPPDA)2](DIPEA) in great detail. 

The title mentions only the lanthanide Yb, while in the synthesis we also see Gd, which is very confusing.

I also do not see a method to confirm the structural analysis of the complex.

Unfortunately, apart from the synthesis, the application of the obtained complex was not demonstrated in the paper, nor was the reproducibility of the described syntheses.

Author Response

Dear reviewer,

        Thank you very much for your meaningful and constructive suggestions. We have made some corrections to our manuscript under consideration of your thoughtful advices. And the responses to your comments have been uploaded in the word file named 'Response to reviewer 2-molecules-2089122' and please see the attachment.

Best wishes

Reviewer 3 Report

Comments and recommendations on molecules-2089122

This paper describes the synthesis and luminescence characterization of two new Ln complexes one of which displays a higher quantum yield and longer lifetime than comparable ones in literature. While being sufficiently new from the compounds as such, the research presented lacks some details in characterization. However, the insufficient presentation of s speculative redox mechanism to explain the  luminescence sensitation mechanism without giving any data as evidence, is not acceptable and the reason for rejection.

Issues that require major revision are indicted (*) as well as reasons for rejection (**):

-          Section 2.1: (*) please indicatethe wavenumbers assigned to the various functional groups also in figure 1.

-          Section 2.3: (*) as to section 3.4, there is additional crystal water contained in [Yb(DPPDA)2](DIPEA). Where in figure 2b can the mass-loss of the crystal water be found? This must be discussed in this section in detail.

-          Section 2.4: (*) figure S8 must have two different y-axes, one with absorbance units and one with luminescence intensity.

Please replace “awfully strong” by “considerably stronger”.

Please indicate, from which literature reference the values for A and B in equation 1 are derived.

Shouldn’t the y axis in figure S14 be labeled with a small Greek letter epsilon? Please revise.

- (*) section 2.4 and 3.2: How do the authors measure absolute quantum yields in solution? Usually, absolute quantum yields are acquired using an integrating sphere. Please give more details in the discussion and the exptl. section.

Section 3.4: (*) there is no logic in giving [Yb(DPPDA)2](DIPEA) as the species found in positive ion mode, as this species is not positively charged. Hence, it should be not detectable in positive ion mode. The correct species must be given along with the correct m/z-value.

-          (**) section 2.5.: it is scientifically incorrect to just speculate on a potential redox process without giving any experimental proof or data. The authors must calculate and compare the related redox potentials of the equilibrium between Yb(III) and Yb(II) and the ligand in oxidized and reduced state under the given conditions to estimate, whether the redox processes they propose is possible. Additionally, it is recommended that the authors should acquire cyclic voltammograms as those should reveal, if their proposed redox mechanism actually occurs or not.

Author Response

Dear reviewer,

        Thank you very much for your meaningful and constructive suggestions. We have made some corrections to our manuscript under consideration of your thoughtful advices. And the responses to your comments have been uploaded in the word file named 'Response to reviewer 3-molecules-2089122' and please see the attachment.

Best wishes

Reviewer 4 Report

The manuscript reports a novel NIR complex that shows a long fluorescence lifetime by suppression of nonradiative decay due to X-H bonds in its coordination sphere. This argument is based on the structure of the complex depicted in Scheme 1 and the analysis with equation (1). However, two nitrogen and two oxygen in a DPPDA dianion would be planer and it is difficult to understand how two DPPDA dianions could form an octacoordinated complex that can prevent small solvent molecules are approaching its inner sphere. In fact, Table 1 shows a larger effect of deuteration by methanol than chloroform, which can be understood by the smaller size of the former than the latter. Two parameters A and B in equation (1) are introduced without any references and it is not possible to understand how the analysis with this equation can lead to the absence of the solvent molecules in the coordination sphere.

The authors propose a nonradiative decay internal redox mechanism instead of ligand-to-metal energy transfer to explain low quantum yield and depict an energy diagram as shown in Figure 5. However, there is no quantitative consideration of this mechanism in the manuscript. The energy level of DPPDA- Yb2+ would be crucial in this scheme and it does not work if this is higher than DPPDA2-* Yb3+. This can be calculated from the standard electrode potential of DPPDA-/DPPDA2- and Yb3+/Yb2+. The latter is known to be around -1.0 V and the former would be estimated from those of similar compounds. 

Line 86~96: The assignment of IR peaks seems a bit confusing: While the peak at 1740cm-1 of DPPDA is assigned to C=O stretching of carboxyl, two peaks at 1590 and 1394cm-1 are also assigned to carboxyl in the ligand. Does this carboxyl actually mean carboxylate of the ligand in a dissociated form? In any case, it seems difficult to understand how the authors could make assignments of the congested peaks in this region and discuss their blue/red shifts by complexation.

Section 2.4: It would be better to note electronic transitions and their energies of Yb3+ and Gd3+ to help understand their difference in fluorescence quantum yields.

Line 191: "the value of ηsens is rather low, which is related to the ineffective energy transfer from the excited energy level of DPPDA to 2F5/2 of Yb3+."  It seems necessary to reconcile this statement with the description in Line 135: "And the significantly diminished visible luminescent intensity of ytterbium complex indicates that the energy absorbed by DPPDA has been plentifully transferred to ytterbium ions."

Author Response

Dear reviewer,

        Thank you very much for your meaningful and constructive suggestions. We have made some corrections to our manuscript under consideration of your thoughtful advices. And the responses to your comments have been uploaded in the word file named 'Response to reviewer 4-molecules-2089122' and please see the attachment.

Best wishes

Round 2

Reviewer 2 Report

comments on the received answers for the article Ren et al. follow:

1. Is the synthesis of Gd complex already known in the literature? if it is, you must add a reference. If the synthesis of Gd complex is done for the first time in this work, it must be analyzed in more detail. The autors cannot compare the new complex (Yb) with the new complex (Gd).

2. I agree that single crystal is extremely difficult to synthesize. The offered structural analyzes are satisfactory.

In Figure 2b. TG curve, please indicate in detail the mass losses for each step, and show the calculation for each step.

Explain cyclic voltammetry in more detail. It is not clear what was intended to be shown with the mentioned method. Cyclic voltammetry results should be presented similarly to these papers:

a) The Journal of Physical Chemistry C 2020 124 (23), 12794-12807 DOI: 10.1021/acs.jpcc.0c02973 

b) Molecules 2022, 27, 3781. https://doi.org/10.3390/molecules27123781

Author Response

Dear reviewer,

        Thank you very much for your meaningful and constructive suggestions. We have made some corrections to our manuscript under consideration of your thoughtful advices. And the responses to your comments have been uploaded in the PDF file named 'Round 2-Response to reviewer 2-molecules-2089122' and please see the attachment.

Best wishes

Reviewer 3 Report

Comments and recommendations on molecules-2089122-v2

The authors invested considerable effort to clarify questions and to enhance the quality of the manuscript which is appreciated. However, two issues still remain to be improved. Therefore, another round of major revision is required. 

-       Section 2.1: (*) “In addition, the infrared absorption peaks …  are attributed to the C=C and C=N stretching of the aromatic ring in ligands”. This statement is still too imprecise. One cannot assign seven different IR-absorption frequencies to vibrations of only two different chemical groups. The authors should consult literature and must assign distinct frequencies to the vibrations of the two different chemical groups.

-       Section 2.3: (*) How can it be that in the new figure 2b, the shape of the curve of the weight loss of the TG curve is exactly the same as in the original submission, but now the range of the weight loss (x-axis) is only between  100 % and 50 %, whereas in the original submission it was between 100 % and 10%? This does not promote the credibility of the data.

Author Response

Dear reviewer,

        Thank you very much for your meaningful and constructive suggestions. We have made some corrections to our manuscript under consideration of your thoughtful advices. And the responses to your comments have been uploaded in the PDF file named 'Round2-Response to reviewer 3-molecules-2089122' and please see the attachment.

Best wishes

Reviewer 4 Report

The issues raised in my previous review report are properly improved with additional measurements and citations and the manuscript is ready for publication.

Author Response

Dear reviewer,

    Thanks for your kind and careful guidance on our work as well as your encouragement and acceptance for our manuscript. Your suggestions benefit a lot and we will be more rigorous and careful in the future.

Best wishes!